# OpenReview forum: "Doubly-Robust LLM-as-a-Judge: Externally Valid Estimation with Imperfect Personas"
_ICLR.cc/2026/Conference — ICLR 2026 Poster_

### Official Review · Reviewer_5tM6 · 2025-10-31

**Soundness:** 3
**Presentation:** 3
**Contribution:** 3
**Rating:** 6
**Confidence:** 4

**Summary:**

This paper solves the problem of external validity in LLM evaluations with limited, biased, or missing human labels. The authors propose a doubly-robust estimator that deals with covariate shift and selective bias. The estimator follows the standard augmented inverse-probability weighted (AIPW) form. The authors further adapt the Riesz loss to estimate the weighting function directly. The experiments validate the approach through three types of datasets: synthetic datasets, semi-synthetic (PRISM) datasets, and real-world (DICES) datasets.

**Strengths:**

- The paper focuses on the timely topic: external validation using LLM. The effectiveness shown by the experiments implies usefulness of this framework in real application.
- The paper establishes the methodologies with high clarity and is grounded in theory.
- The experiment design is comprehensive. The PSF allows systematic manipulation of covariate shift, selection bias, and persona quality and provides transparent benchmarks for future work.

**Weaknesses:**

- Though repeated for times, it is still not clear what this doubly-robustness means for the LLM-as-a-judge framework. I would really appreciate it if the authors can clarify this (see details in questions).
- Conceptual novelty is over-claimed. The estimator is essentially the AIPW/doubly-robust estimator widely studied in missing-data and causal-inference literature.

**Questions:**

- (Related to the first weakness) What does these conditions mean (quoting from abstract): "either (i) a model trained to predict human ratings using persona ratings and source data observed under sampling bias, or (ii) a reweighting model that corrects for sampling bias is of sufficient quality"? Can the authors explain more on this? What could be an example where the conditions are violated?
- Why coverage is a fair metric for comparing the methods in the experiment? How are the confidence intervals constructed by the baselines?

---

> ### Author Response · Authors · 2025-11-26
> **Response -- Conceptual Novelty**
>
> We would like to thank the reviewer for their helpful feedback on our work. The reviewer raises several important points surrounding the presentation of our technical framework and results, which we carefully address below. First, in this comment, we respond to the reviewer's concerns surrounding conceptual novelty.
>
> > Conceptual novelty is over-claimed. The estimator is essentially the AIPW/doubly-robust estimator widely studied in missing-data and causal-inference literature.
>
> We appreciate this observation. While the underlying mathematical building blocks (doubly-robust estimation, Riesz loss) are well-established in causal inference, we view our contribution as unifying these rigorous statistical tools with the emerging, heuristic-heavy field of GenAI evaluation.
>
> **1. Problem Formulation.** The core novelty of our work lies in recognizing that GenAI system quality estimation can be cast as a causal inference problem under covariate shift and selection bias. This connection is currently absent from the GenAI evaluation literature, where existing approaches rely on ad-hoc methods we study as baselines (Persona-Based Estimator, Persona-Augmented Regression). These baselines lack theoretical justification, and, as our experiments illustrate, produce severe bias and miscoverage. Even state-of-the-art statistical tools (e.g., PPI++ [1], RePPI [2]) do not jointly account for selection bias and covariate shift; for example, when raters selectively drop out based on content difficulty while also differing from the target deployment distribution. As a result, our experiments show these approaches remain brittle to realistic evaluation challenges.
>
> More generally, despite several years of widespread deployment of GenAI systems, existing literature has not made this foundational connection between DR methodology and GenAI evaluation explicit. Bridging these domains required carefully formalizing how different patterns of data missingness arise in a GenAI evaluation setting (e.g., via rater dropout, non-compliance, and rater/content shift), and how persona ratings can be used as surrogates to improve inference. Both represent non-trivial conceptual steps that unlock a rich toolkit of methods for a practically important problem.
>
> **2. Technical Contributions.** The reviewer is correct that doubly-robust estimation and Riesz losses are established techniques. Applying these tools to our setting required non-trivial technical extensions that remain novel within the causal inference literature. Specifically, our setting requires handling the joint presence of non-compliance and covariate shift within general M-estimation; a combination not addressed by prior work. For instance, Zheng et al. [3] consider the estimation of an expected outcome under both non-compliance and covariate shift. However, the work (a) does not apply to general M-estimation and (b) does not show how to leverage the Riesz loss to estimate complicated nuisances (e.g. products of inverse probabilities of non-compliance and likelihood ratios). Similarly, Chernozhukov et al. [4] considers the problem of estimating general linear functionals of a regression using automatic de-biasing methods based on the Riesz loss. However, the results (a) do not allow for non-compliance and (b) again do not hold for general M-estimation problems. We view our contributions as a natural generalization of both of these works. These extensions, while grounded in established theory, represent meaningful technical contributions that enable our framework.
>
> This technical generalization to M-estimation problems is practically significant. In a newly added Table 1, we outline the numerous system quality parameters that can now be estimated as part of our new technical framework.
>
> **Summary**
>
> We view our work as similar in spirit to influential papers like PPI++ [1] and Predictions as Surrogates [2], which also adapt established statistical techniques to emerging problems in AI evaluation. We believe our formalization opens a rich space of follow-up work, such as handling unmeasured confounding or addressing positivity violations in high-dimensional embeddings.
>
> **References:**
>
> [1] Angelopoulos, Duchi, Zrnic, PPI++, 2024, https://arxiv.org/abs/2311.01453
>
> [2] Ji, Lei, Zrnic, Predictions as Surrogates, 2025, https://arxiv.org/abs/2501.09731
>
> [3]  Zheng et al., Efficient Generalization and Transportation, 2024, https://arxiv.org/pdf/2302.00092
>
> [4] Chernozhukov et al., Automatic Debiased Machine Learning for Covariate Shifts, 2023, https://arxiv.org/abs/2307.04527

---

> ### Author Response · Authors · 2025-11-26
> **Response -- Double-Robustness Property**
>
> > Though repeated for times, it is still not clear what this doubly-robustness means for the LLM-as-a-judge framework…What does these conditions mean (quoting from abstract): "either (i) a model trained to predict human ratings using persona ratings and source data observed under sampling bias, or (ii) a reweighting model that corrects for sampling bias is of sufficient quality"? Can the authors explain more on this? What could be an example where the conditions are violated?
>
> Thank you for raising this important point. We're happy to clarify the double-robustness property and its connection to LLM-as-a-judge.
>
> **Intuition:** Our estimator combines two models: a model that predicts human ratings and one that reweights data to correct for sampling bias. At a high level, "doubly-robust" property means that if either model is sufficiently accurate (converges at a fast enough rate), our overall estimate remains valid and our confidence intervals maintain proper coverage. In contrast, if both models fail to converge at a sufficiently fast rate, then our method may obtain invalid coverage.
>
> **When does this fail? A counter-example:** Suppose we fit both the human rating model and the reweighting model using a simple parametric model: e.g., linear regression for predicting human ratings and logistic regression for the reweighting function. If the true relationships were highly non-linear (e.g., ratings depend on complex interactions between content features and rater demographics), both models would be misspecified, leading to large errors in both components. When both fail simultaneously, the double-robustness condition is violated (see formal description below), and our estimator fails to provide valid coverage. In practice, we avoid this failure by using flexible models (neural networks or gradient-boosted trees) for both components, as is standard in the doubly-robust estimation literature [1].
>
>
>
> **Connection to LLM-as-a-Judge:** In our framework, we leverage the LLM-as-a-judge with persona ratings to make the double-robustness property easier to satisfy in practice. In our framework, the outcome model predicts human ratings using both content/rater features (W) and the LLM persona rating ($\hat{Y}$): $\mu = E[Y | W, \hat{Y}].$ When the persona rating is highly correlated with the true human rating, it becomes easier to learn an accurate outcome model $\hat{\mu}$, even when we have a hard time learning an accurate reweighting model $\alpha$. This triggers the double-robustness property: high quality persona ratings improve the human rating model, which in turn propagates to higher quality downstream estimates.  Figure 3 (top row) illustrates this phenomenon in action, because we observe improved coverage in doubly-robust estimates when the correlation between persona ratings and human ratings is sufficiently high (e.g., $\rho>0.6$).
>
> **Formal Details:** For completeness, we provide the technical statement of the double-robustness condition. Let $\mu_0 = E[Y \mid W, \hat{Y}]$ denote the outcome regression and $\alpha_0(W,C) = \frac{C\omega_0(W)}{\pi_0(W)}$ denote the reweighting function. When learning the estimated functions $\hat{\mu}$ and $\hat{\alpha}$, the double-robustness condition requires: $$\|\|\hat{\alpha} - \alpha_0\|\| \cdot \|\|\hat{\mu} - \mu_0\|\| = o_{\mathbb{P}}(N_t^{-1/2}),$$ where both norms denote the L_2 norm, and $N_t$ is the number of target samples. In plain language, the product of estimation errors must decay at a parametric rate. This is satisfied if both models converge at ${N_t}^{-1/4}$ rates individually (typical for flexible nonparametric estimators), because their product achieves the requisite ${N_t}^{-1/2}$ rate.
>
> **Connecting to High-Level Language:** Finally, we connect this back to the phrasing from our abstract that caused some confusion:
> - *"A model trained to predict human ratings using persona ratings and source data"* refers to the learned outcome regression $\hat{\mu}$.
> - *"A reweighting model that corrects for sampling bias"* refers to the learned reweighting function $\hat{\alpha}$.
> - *"Of sufficient quality"* means the product of errors decays at the $N_t^{-1/2}$ rate shown above.
>
> To help readers understand these concepts, we have substantially expanded our discussion of the double-robustness property in Section 3.2 (see Equation 3 and surrounding text, highlighted in blue), connecting our plain-language description to the formal condition.
>
> **References:**
>
> [1] Edward H. Kennedy, Semiparametric doubly robust targeted double machine learning: a review, 2022, https://arxiv.org/abs/2203.06469

---

> ### Author Response · Authors · 2025-11-26
> **Response -- Empirical Validation**
>
> Finally, the reviewer raises a question regarding our empirical validation:
>
> > Why is coverage a fair metric for comparing the methods in the experiment? How are the confidence intervals constructed by the baselines?
>
> In line with prior work validating statistical estimators [1,2], we report three complementary metrics: coverage, bias, and interval width. Coverage is featured prominently in Figure 3 because *invalid coverage* could lead practitioners to incorrect conclusions about system performance and thus deploy flawed systems. We also report detailed breakdowns of interval width and bias in Figures 11-13.
>
> **Why These Metrics?**
> Reporting all three metrics enables us to triangulate multiple dimensions of estimator quality:
> - **Coverage**, which measures the empirical probability (across repeated trials) that the confidence interval covers the true parameter $\Pr(\theta_t \in CI)$, helps us understand if we are likely to make an invalid statistical assessment of a system’s performance, after accounting for sampling uncertainty.
> - **Bias**, which measures the absolute deviation from the target parameter $|\theta_t - \hat{\theta}_t|$, helps us understand systematic discrepancies between the estimate and target parameter. High bias indicates that we are likely to overestimate or underestimate some property of the target system (e.g., helpfulness, toxicity) when obtaining a system quality estimate.
> - **Interval Width**, which measures the size of the confidence interval, helps us understand the informativeness of the system quality estimate. Reporting this metric is important because one could trivially satisfy coverage by reporting a very wide confidence interval. But this wide confidence interval would not convey meaningful information about the true performance of the system.
>
> Across settings, we find our method has much higher coverage and lower bias than baselines, while maintaining similar or only slightly wider interval widths. We do observe some methods with smaller intervals (e.g., Persona-Based). But because of the high bias and low coverage, the estimator is producing high confidence, incorrect system quality estimates. In contrast, our DR estimator has slightly wider intervals because it appropriately reflects statistical uncertainty attributable to evaluation sampling bias.
>
>
> **Confidence Interval Computation:** For PPI++ [1] and RePPI [2], we use the authors' provided implementations to ensure fair comparison. For simpler baselines (Source Mean, Persona-Based), we use closed-form expressions based on asymptotic variance. Code for all confidence interval calculations can be found in the estimators.py file attached in the supplementary material.
>
> **References:**
>
> [1] Angelopoulos, Duchi, Zrnic, PPI++, 2024, https://arxiv.org/abs/2311.01453
>
> [2] Ji, Lei, Zrnic, Predictions as Surrogates, 2025, https://arxiv.org/abs/2501.09731

---

### Official Review · Reviewer_hCKA · 2025-10-31

**Soundness:** 4
**Presentation:** 4
**Contribution:** 3
**Rating:** 8
**Confidence:** 3

**Summary:**

This paper studies the external validity of LLM-as-a-judge when there are covariate shift and selection bias in human ratings. It proposes a doubly robust estimator that combines the LLM-data-augmented regression estimator and the reweighting estimator, and proves the asymptotic normality of the estimator. In the numerical experiments, it proposes a persona simulation framework to vary the degrees of covariate shift, selection bias, and persona quality. Experiment results show reduced bias, good confidence interval coverage and confidence interval width.

**Strengths:**

1. The research problem of external validity in LLM-as-a-judge is practically meaningful.
2. The paper is well written.
3. The proposed doubly robust estimator jointly handles covariate shift and selection bias. The theoretical results are solid.
4. The experiment results are compelling, and demonstrate the advantages of the proposed estimator over baselines.

**Weaknesses:**

1. While the theory is solid, the core mathematical ideas involved (doubly robust estimation, Riesz loss) are standard in the literature.
2. The validity of the method hinges on the assumption that the covariates $W=(X,V)$ captures all factors that influence the rating $Y$ and the probability of completing a rating $C$. This can be a strong assumption in practice.
3. In the numerical experiments, the data points are projected to 15 dimensions (line 323). It is questionable if this 15-dimensional vector is sufficiently rich to satisfy the assumption mentioned in the previous point, which leads to a potential mismatch between theory and experiment.

**Questions:**

1. When the data points are projected to 15 dimensions in the experiment, how are these 15 dimensions selected to retain some predictive signal?

---

> ### Author Response · Authors · 2025-11-26
> **Response -- Conditional Ignorability Assumption**
>
> We thank the reviewer for reading our paper and providing feedback. We are glad that the reviewer finds our theoretical framework solid, our problem practically meaningful, and experimental results compelling. We also appreciate the thoughtful feedback regarding the conditional ignorability assumption and the dimensionality reduction step performed in experiments. We address these points below, starting in this comment with the conditional ignorability concerns.
>
> > The validity of the method hinges on the assumption that the covariates capture all factors that influence the rating and the probability of completing a rating . This can be a strong assumption in practice.
>
> Thank you for raising this point. We agree that this conditional ignorability assumption may indeed be unrealistic in some evaluation settings.  Our main goal in this work is to cast the problem of system quality estimation under evaluation sampling bias as an explicitly causal problem. This provides conceptual clarity and enables us to leverage statistical tools for tackling the problem head-on. Having established our framework, practitioners can explore multiple avenues to relax this assumption:
>
> 1. First, one could attempt to gather a richer set of covariates for those involved in the study. If one is just using open source data (as in our case), this approach may be unreasonable.
> 2. Second, and more realistically, one could use tools from the sensitivity analysis literature ([1] being the most relevant example) to understand robustness of our confidence intervals to unobserved confounding. In particular, one could compute "robustness values" to assess the strength of unobserved confounding necessary to overturn any discovery made (e.g. validity of constructed confidence intervals). These results may need to be slightly modified/extended to hold in our setting.
>
> We view sensitivity analysis as valuable future work that would naturally extend our framework. We also note that prior related work (e.g., [2,3]) implicitly relies on similar ignorability conditions, but leaves these assumptions unstated and thus difficult to diagnose. By making the causal structure explicit, our framework enables practitioners to reason about when these assumptions may be violated and import existing sensitivity analysis tools from causal inference to relax them carefully.
>
>
> **References:**
>
> [1] Chernozhukov et al., Long Story Short: Omitted Variable Bias in Causal Machine Learning, 2021, https://arxiv.org/abs/2112.13398
>
> [2] Angelopoulos, Duchi, Zrnic, PPI++, 2024, https://arxiv.org/abs/2311.01453
>
> [3] Ji, Lei, Zrnic, Predictions as Surrogates: Revisiting Surrogate Outcomes in the Age of AI, 2025, https://arxiv.org/abs/2501.09731

---

> ### Author Response · Authors · 2025-11-26
> **Response -- Dimension Reduction Step in Experiments**
>
> Related to the conditional ignorability concern discussed above, the reviewer also probes our use of low-dimensional embedding projections in our experiments.
>
> > Data points are projected to 15 dimensions (line 323). It is questionable if this 15-dimensional vector is sufficiently rich to satisfy the assumption mentioned in the previous point. How are these 15 dimensions selected to retain some predictive signal?
>
> We thank the reviewer for highlighting this trade-off between satisfying conditional ignorability via rich covariates and practical estimation challenges that arise when learning nuisance functions (i.e., avoiding overfitting).
>
> During the experiment design, we aimed to identify an embedding sub-space with sufficient predictive signal, while also keeping dimensionality low enough to train stable nuisance models. In particular, we selected $D=15$ by varying the number of dimensions recovered from UMAP, $D \in [1, 384]$, then measuring the average correlation between projected features and our downstream variables: human ratings ($Y$) and source/target membership ($A$). As illustrated in the **newly added Figure 7 in Appendix E.4**, we observed an "elbow" emerge at $D=15$. This indicated that adding further dimensions yielded diminishing returns in predictive signal.
>
> The reviewer helpfully notes that if this projection discards features that act as confounders, the conditional ignorability assumption would be violated. While this is possible in principle, our empirical results illustrate that violations of the conditional ignorability assumption did not pose a significant practical issue. In particular, we observe nominal 95% coverage across a wide range of empirical conditions (see Figure 3). We view this as a necessary practical tradeoff. Including the full high-dimensional embedding would likely cause the nuisance estimators to overfit (high variance), which may similarly degrade coverage.
>
> We will add a discussion of this dimensionality-overfitting tradeoff and expand on the feature selection diagnostic (Figure 7) in the final version of the paper to help practitioners navigate similar issues in the future.

---

> ### Author Response · Authors · 2025-11-26
> **Response -- Novelty**
>
> Finally, the reviewer raises questions surrounding the novelty of the core statistical methods:
>
> > While the theory is solid, the core mathematical ideas involved (doubly robust estimation, Riesz loss) are standard in the literature.
>
> We appreciate this important point, which was also raised by Reviewer 5tM6. While doubly-robust estimation and Riesz losses are indeed established techniques, our work makes several novel contributions:
>
> **Novel Problem Formulation:**
> - We are the first to formalize GenAI system quality estimation as a causal inference problem with missing data, whereby *simultaneous* covariate shift and selection bias can impact human ratings. This moves beyond existing statistical frameworks used for system quality estimation (PPI++, RePPI) that assume data is missing completely at random (MCAR) or drawn i.i.d. from source and target distributions.
> - We introduce a principled framework for incorporating LLM-generated persona ratings as informative yet imperfect proxies for human ratings in this motivating setting.
>
> **Technical Contributions:**
> - We extend doubly-robust M-estimation theory (Chernozhukov et al., 2023 [1]) to settings with joint covariate shift and selection bias. This enables inference on broader classes of parameters (variance, quantiles) that are relevant to system quality estimation tasks.
> - We demonstrate how to apply Riesz losses to high-dimensional text spaces using embedding projections, addressing practical challenges not solved in prior work.
> - We introduce the Persona Simulation Framework (PSF), a reusable experimental infrastructure for validating estimators under controlled evaluation sampling bias.
>
> We provide a more detailed discussion of these novelty dimensions in our response to Reviewer 5tM6. To summarize: while we build on established statistical techniques, our contribution lies in (i) identifying and formalizing a critical gap in GenAI evaluation methodology, (ii) adapting and extending existing theory to address this gap, and (iii) demonstrating practical viability through systematic experiments.
>
> We thank the reviewer for their detailed attention to our work and for highlighting the connection between theoretical assumptions and experimental validation. We hope the additions to the paper (including Figure 7 in Appendix E.4) help clarify these points.
>
> [1] Chernozhukov et al., Automatic Debiased Machine Learning for Covariate Shifts, 2023, https://arxiv.org/abs/2307.04527

---

### Official Review · Reviewer_radJ · 2025-11-02

**Soundness:** 3
**Presentation:** 2
**Contribution:** 3
**Rating:** 6
**Confidence:** 2

**Summary:**

This paper studies how to estimate the external validity of human evaluations when there is a distribution shift between lab raters and real-world raters. The authors propose a doubly robust estimation framework that accounts for both covariate shift and selection bias, aiming to make lab-based evaluation results more representative of deployment settings. Experiments on three datasets show improved estimation quality over baselines.

**Strengths:**

- The paper rigorously formulates the problem.
- The motivation is strong: external validity is a real issue for GenAI evaluation, especially in high-stakes domains like healthcare.
- The proposed method is well motivated, and the experiments demonstrate performance gains.

**Weaknesses:**

- The writing can be improved for clarity and flow.
    - Figure 1 is difficult to follow in the introduction and may be better placed in the experiments section.
    - The paper is notation-heavy, which may hinder readability; simplifying or deferring some notation might help.
- All three datasets are somewhat synthetic. While this is understandable, the lack of real-world data weakens the external claims.It would be helpful to include qualitative insights into how varying levels of covariate shift manifest in the datasets.

**Questions:**

- Q1: Covariate shift is controlled in the embedding space. Could you share qualitative examples or interpretations of how source and target distributions differ? Which factor—content (V) or rater features (X)—contributes more to the observed shift?

- Q2: In your motivating example where students label source data and experts label target data, the raters have very different characteristics. Have you considered manually constructing such a scenario to validate model robustness under extreme rater mismatch?

---

> ### Author Response · Authors · 2025-11-26
> **Response -- Intuition for Empirical Setup**
>
> We thank the reviewer for their thoughtful feedback on our work. We appreciate that the reviewer finds the external validity problem practically meaningful, and believes that our doubly-robust approach is well motivated.  In this comment, we respond to one of the reviewer's concerns about the work, which involves helping readers build intuition for empirical setup, through comparative breakdowns and qualitative examples:
>
> > All three datasets are somewhat synthetic…It would be helpful to include qualitative insights into how varying levels of covariate shift manifest in the datasets. Could you share qualitative examples or interpretations of how source and target distributions differ? Which factor—content ($V$) or rater features ($X$)—contributes more to the observed shift?
>
> We agree that providing qualitative examples and joint analysis of both shift factors would strengthen the readers’ understanding of our setup. Our newly added Figure 6 in Appendix E.4 provides intuition for how covariate shift varies across experimental conditions in the DICES dataset.
>
> **Joint Analysis:** Panel (b) of Figure 6 visualizes the Sinkhorn Distance (color) as a function of content overlap ($\zeta$, x-axis) and rater shift magnitude (y-axis). The vertical orientation of the contour lines clearly illustrates that content features ($V$) drive the covariate shift more than rater features ($X$). As we move up the y-axis (increasing rater shift), the color (total shift) remains largely constant. Conversely, as we move right along the x-axis (decreasing content overlap), the color shifts from purple to yellow, indicating an increase in distribution shift. We observe two causes for the larger contribution of content features: first, there are fewer rater features ($D_x=3$) than content features ($D_v=15$) in the DICES dataset. Second, because we constructed DICES to resemble realistic demographic shifts (Uniform vs. Census), the maximum magnitude of rater shift is naturally bounded compared to the larger semantic shifts possible in text embeddings.
>
> **Rater Shift Analysis:** Panel (a) breaks down this finding further, by providing a qualitative example of how the rater mixture parameter affects the probability of sampling two different subgroups. When the rater mixture is 0, we observe a high probability of sampling $X_1$ = (Asian, Gen Z, College degree or higher), and a low probability of sampling $X_2$ = (White, Gen X+1, High school or below). However, when the rater mixture is 0.75 or higher, this inverts such that sampling $X_2$ is more likely than sampling $X_1$.
>
> **Content Shift Analysis:**  Panel (c) illustrates how content distributions provided in synthetic datasets vary in parameter $\zeta$. The top comment (labeled with *Religious and Misinformation* harms) is a member of the target distribution. As $\zeta \rightarrow 1$ (more source/target overlap), this content is more likely to be sampled into the source distribution during the synthetic dataset construction process. This makes the magnitude of covariate shift in the sample smaller, as illustrated by the smaller Sinkhorn Distance shown in purple in the contour plot. Conversely, the bottom comment (labeled as *Regulated Goods*) is a member of the source distribution. As $\zeta \rightarrow 0$ (less overlap), this content has a lower probability of being removed from the source sample (since source samples are removed to maintain a fixed source/target sample size). The net effect of reducing $\zeta$ is a higher Sinkhorn Distance, as illustrated by the yellow regions of the plot.

---

> ### Author Response · Authors · 2025-11-26
> **Response -- Estimation under Severe Rater Mismatch**
>
> > Q2: In your motivating example where students label source data and experts label target data, the raters have very different characteristics. Have you considered manually constructing such a scenario to validate model robustness under extreme rater mismatch?
>
> Thank you for this excellent suggestion. Our experimental setup systematically probes robustness to covariate shift, though we focused on ranges of realistic rater demographic shifts rather than the extreme rater mismatch scenario suggested in the comment. In our analysis, the second row of Figure 3 is particularly relevant to this question: under what magnitude of covariate shift does our approach fail to produce valid coverage? For PRISM, coverage remains valid when $\Delta \leq 2$, but degrades above that threshold. This suggests that extreme rater mismatch scenarios (e.g., disjoint groups of students vs. experts) would likely exceed this threshold, causing coverage failures for our method and all baselines.
>
> Prior work in causal inference has explored such failures in detail, under the name of so-called “positivity violations” [1]. Such work often emphasizes a theoretical limit to the reliability of statistical conclusions that can be drawn under extreme overlap violations. Characterizing the impact of such overlap violations for GenAI system quality estimation would be an interesting line of future work.
>
> **Revising the motivating example.** Your question helped us realize our opening example may set misleading expectations about the magnitude of covariate shift that our framework addresses. We've revised the motivating example (see changes in blue, lines 44-46) to better align with the moderate shift magnitudes we empirically validate. We hope that this will create stronger alignment between how we motivate our framework and how it is validated.
>
> [1] D'Amour et al., Overlap in observational studies with high-dimensional covariates, 2021, https://www.sciencedirect.com/science/article/pii/S0304407620302694

---

> ### Author Response · Authors · 2025-11-26
> **Response -- Presentation**
>
> Additionally, the reviewer helpfully points out several areas where the presentation of our work can be improved.
>
> > The paper is notation-heavy, which may hinder readability; simplifying or deferring some notation might help.
>
> We agree that the notation can be quite heavy in places - particularly in Section 3. We have taken a careful pass through Section 3 to add plain language explanations alongside the formal definitions. We especially emphasized providing intuition for the double-robustness property, since other reviewers also asked for greater clarity on this concept. Unfortunately, given the statistical setup, we found it quite difficult to fully eliminate our reliance on notation without sacrificing clarity. We note that several of our most related works (e.g., PPI++ [1]; RePPI [2]), communicate technical ideas at a comparable level of mathematical detail. We do care deeply about the accessibility of work. We plan to write a clear and accessible blog post and project README to ensure the work is legible to an evaluation-focused audience.
>
> [1] Angelopoulos, Duchi, Zrnic, PPI++: Efficient Prediction-Powered Inference, 2024
>
> [2] Ji, Lei, Zrnic, Predictions as Surrogates: Revisiting Surrogate Outcomes in the Age of AI, 2025
>
> &nbsp;
> > Figure 1 is difficult to follow in the introduction and may be better placed in the experiments section.
>
> Thank you for this helpful suggestion. In response, we've substantially expanded Figure 1's caption and surrounding text to improve accessibility. The caption now explains what  each baseline method does and clearly states our main finding (valid coverage + low bias of our DR approach). In the introduction, we also added text connecting this preview to the comprehensive results in Section 4. While we thoroughly considered moving Figure 1 to experiments, we believe the improved explanation makes it more accessible while serving its purpose: quickly communicating to statistical readers the improved properties of our estimator (coverage and bias). We hope the planned blog post will further help non-statistical audiences engage with this work.
>
> We hope these revisions and clarifications address your concerns. We appreciate your feedback and believe the changes we have made will also help other readers better-understand our framework.

---

### Author Response · Authors · 2025-11-26
**General Response to Reviewers**

We thank the reviewers for their thorough assessment and constructive feedback. We appreciate that all reviewers (5tM6, hCKA, radJ) find the problem of external validity addressed by our framework timely and meaningful, and our technical approach sound. The reviewers also raise several important concerns, which we have addressed via revisions to the manuscript and detailed individual replies:

**Novelty (Reviewers hCKA, 5tM6):** Two reviewers noted that doubly-robust estimation and Riesz losses are established causal inference techniques. While true, our contribution lies in bridging the gap between high-level calls for external validity in GenAI evaluation, made in recent position papers [1,2], and a rigorous statistical framework that operationalizes these challenges in practice. This formalization matters practically. As Figure 3 demonstrates, state-of-the-art statistical methods (PPI++, RePPI) fail to maintain valid coverage under realistic evaluation conditions because they don't account for simultaneous covariate shift and selection bias.

Our novelty operates at two levels:
- (i) *Problem formulation* - We formalize how data missingness arises in GenAI evaluation (rater dropout, demographic shift) and how imperfect LLM persona ratings can be incorporated as surrogates. This unlocks a rich toolkit currently unavailable to practitioners.
- (ii) *Technical contributions* - We generalize prior doubly-robust frameworks [3,4] to M-estimation settings with simultaneous covariate shift and selection bias. This enables inference on evaluation parameters beyond means (variance, quantiles; see newly added Table 1), which is practically important for system quality estimation.

**Experimental Setup (All reviewers):** Reviewers requested clarification on how covariate shift manifests in our experimental setup (radJ), our dimensionality reduction approach (hCKA), and evaluation metrics (5tM6). In response, we have added:
- *Figure 6 (Appendix E.4):* A three-panel visualization showing content features drive covariate shift more than rater features in DICES; includes qualitative examples
- *Figure 7 (Appendix E.4):* A correlation diagnostic plot justifying D=15 dimensionality via an “elbow” heuristic
- *Metric justification:* We report coverage, bias, and interval width to triangulate estimator quality across multiple dimensions (response to 5tM6).

**Presentation (Reviewers radJ, 5tM6):** Reviewer 5tM6 requested clarification of how the double-robustness condition applies in our setting. We substantially expanded Section 3.2 (Equation 3, highlighted in blue) with: (i) intuitive explanation, (ii) the formal product-of-errors condition, and (iii) connection to how persona ratings connect to this condition. Reviewer radJ suggested improving Figure 1 and reducing notation; we expanded Figure 1's caption and added plain-language explanations throughout Section 3.

**Summary of Revisions to Manuscript:**
- Figure 1: Expanded caption explaining baselines and main findings
- Introduction (lines 44-46): Revised motivating example to reflect moderate shift magnitudes
- Table 1: Examples of M-estimation parameters (variance, quantiles) now addressable
- Section 3.2: Expanded double-robustness explanation (Equation 3 + surrounding text)
- Figures 6-7 (Appendix E.4): Added covariate shift visualization and dimensionality reduction diagnostic plots

In summary, our work establishes the first principled statistical framework for GenAI evaluation under sampling bias. While we build on established causal inference tools, the connection to external validity in GenAI evaluation is novel and practically important. Our problem formulation, theoretical framework, and Persona Simulation Framework for estimator validation provide a foundation for future methodological work addressing external validity challenges.

We appreciate this opportunity to strengthen the paper and look forward to continued discussion.


**References:**

[1] Weidinger et al., Toward an evaluation science for generative ai systems, 2025, https://arxiv.org/abs/2503.05336

[2] Salaudeen et al., Measurement to meaning: A validity centered framework for ai evaluation, 2025, https://arxiv.org/abs/2505.10573

[3] Chernozhukov et al., Automatic Debiased Machine Learning for Covariate Shifts, 2023, https://arxiv.org/abs/2307.04527

[4] Zheng et al., Efficient Generalization and Transportation, 2024, https://arxiv.org/pdf/2302.00092

---

### Meta-Review · Area_Chair_qQV3 · 2025-12-27

**Summary:**

This paper focuses on the external validity of LLM-as-a-judge under covariate shift and selection bias in human raters. The authors propose a doubly robust estimator that combines an LLM-data-augmented regression estimator with a reweighting estimator and prove the asymptotic normality of the resulting estimator. Reviewers raised concerns regarding the problem formulation, methodology, experimental setup for validating the motivation, and presentation. The authors addressed these concerns by clarifying the formulation, adding persuasive experiments, and revising the manuscript accordingly. Overall, this is a good work with uniformly positive reviews. I would like to recommend a further revision in response to the reviewers’ comments.

**Reviewer Concerns:**

The authors have addressed most concerns regarding the experiments and presentation:

- by providing qualitative analyses with examples illustrating how covariate shift varies across experimental conditions;

- by clarifying experimental details, such as metric design and the selection of dimension number.

However, for this theory-applied work in LLM systems, reviewers also noted issues with notation, potential overclaims regarding novelty, and strong underlying assumptions. I recommend a careful review of the notation and a strengthening of the contribution through a clearer formulation of the problem in the context of LLMs.

**Reviewer Scores:**

The original reviews have already supported acceptance of the paper.

---

### Decision · Program_Chairs · 2026-01-26

Accept (Poster)